# Association between Yoga Participation and Arterial Stiffness: A Cross-Sectional Study

**DOI:** 10.3390/ijerph20105852

**Published:** 2023-05-17

**Authors:** Tilak Raj, Catherine A. Elliot, Lee Stoner, Simon Higgins, Craig Paterson, Michael J. Hamlin

**Affiliations:** 1Department of Tourism, Sport and Society, Lincoln University, Christchurch 7647, New Zealand; tilak@touchnz.co.nz (T.R.); catherine.elliot@lincoln.ac.nz (C.A.E.); 2Department of Exercise and Sport Science, University of North Carolina at Chapel Hill, Chapel Hill, NC 27599, USA; stonerl@email.unc.edu (L.S.); higginss@unc.edu (S.H.); craigpat@unc.edu (C.P.)

**Keywords:** arterial stiffness, blood pressure, heart disease, physical activity, pulse wave velocity

## Abstract

Background: Yoga may help adults of all fitness levels increase their physical activity and decrease their cardiovascular disease risk. Aim: To determine if arterial stiffness is lower (beneficial) in yoga versus non-yoga participants. Method: This cross-sectional study included 202 yoga (48.4 + 14.1 years, 81% female) and 181 (42.8 + 14.1 years, 44% female) non-yoga participants. The primary outcome was carotid–femoral pulse wave velocity (cfPWV). The two groups were compared using analysis of covariance with adjustments for demographic (age and sex), hemodynamic (mean arterial pressure and heart rate), lifestyle (physical activity levels, sedentary behaviour, smoking status and perceived stress score) and cardiometabolic (waist-to-hip ratio, total cholesterol and fasting glucose) factors. Results: Following adjustments, cfPWV was significantly lower in yoga compared to non-yoga participants with a mean difference: −0.28 m.s^−1^, (95% CI = −0.55 to 0.08). Conclusion: At a population level, yoga participation may assist with decreasing the risk of cardiovascular disease in adults.

## 1. Introduction

Arterial stiffening refers to a diminished capacity of an artery to expand and contract in response to pressure changes, and is associated with structural and functional changes in the blood vessel, resulting in decreased vascular distensibility [1], and is a major cardiovascular disease (CVD) risk factor. A reduced ability of the artery to absorb pressure increases the risk of damage to the arterial wall and heightens the risk of transmitting potentially damaging pressure waves along the blood vessels to the body’s organs. CVD risk may be mitigated through regular engagement in physical activity [2]. Physical activity is most beneficial if performed at a moderate-to-vigorous intensity [2]. However, regular participation in yoga, generally a low-intensity exercise encompassing stretching and relaxation techniques, is also reported to be associated with lower (beneficial) arterial stiffness.

Prior work has shown that yoga (compared to brisk walking), practiced for 1 h per day, 6 days per week over 6 or 12 weeks, significantly reduced systolic blood pressure [3] and arterial stiffness [4] in elderly hypertensive participants. It is thought that yoga itself (through controlled breathing and relaxation) enables a reduction in the sympathetic innervation of the smooth muscle at the blood vessel [5], while others have postulated that the regular stretching that occurs with yoga may cause increased release and subsequent enhanced bioavailability of nitric oxide, causing endothelium-dependent decreased vasoconstrictor tone [6]. However, such acute studies rarely account for lifestyle factors (for example, leisure-time physical activity) or underlying metabolic conditions (e.g., pre-diabetes), and whether similar positive cardiovascular effects are evident in a healthy population practicing yoga is yet to be determined. Therefore, a study is needed to investigate whether participating in regular yoga sessions is beneficial for lowering arterial stiffness in a sample of otherwise healthy adults after accounting for major lifestyle and cardiometabolic confounders. 

The objective of this study is to determine whether arterial stiffness is lower (beneficial) in yoga versus non-yoga participants. Addressing this objective is important, considering that yoga has emerged as a popular component of the Western fitness industry [7]. This study addressed gaps in the literature by examining the association between long-term use of yoga and arterial stiffness in a non-hypertensive otherwise healthy adult population after accounting for potential CVD confounders.

## 2. Materials and Methods

This cross-sectional study was carried out in accordance with STROBE (Strengthening the Reporting of Observational Studies in Epidemiology) guidelines [8]. The study was conducted in accordance with the Declaration of Helsinki, and the protocol was approved by the University Human Ethics Committee (reference 2018-42) with all participants providing written informed consent. 

### 2.1. Study Design and Overview

This cross-sectional study was conducted at a university in the Canterbury region of New Zealand between January and September 2019. Participants reported for a single one-hour testing session between 5:00 a.m. to 11:30 a.m., having fasted for 12 h, abstaining from strenuous physical activity and alcohol for 24 h, and refraining from supplement intake that morning. Hamstring flexibility of participants was measured in a seated position in front of a Flex-Tester© box (Novel Products, Inc.; Rockton, IL, USA). The participants performed the hamstring flexibility test without any warm-up. Participants were instructed to keep their knees fully extended in front, and the soles of their feet placed against the box. Participants then placed one hand flat on the top of the other hand and pushed the measuring tab of the box in a slow and controlled movement as far as possible and continued breathing as they held their furthest position for two seconds. There was a 2 min rest between each of the three attempts, during which participants could choose to either stand up or sit down and drink water. The best attempt was used in the analysis. The sit-and-reach test used in this research has been shown to be a valid and reliable measure of hamstring flexibility (Jackson and Baker 1986) [9]. Following 20 min quiet rest, arterial stiffness and hemodynamic measurements were collected, followed by a finger prick to collect a capillary blood sample for cardiometabolic measures. Participants then completed a series of questionnaires to collect demographic and lifestyle behaviour data.

### 2.2. Participants

We recruited healthy men and women between the ages of 18 and 85 years using mass emailing, social media and flyers in yoga studios. Participants were excluded if they were taking medications to control blood pressure, cholesterol or blood sugar, or had a known, uncontrolled cardiometabolic disorder. Participants were separated into yoga (practiced for at least once/week for ≥12 weeks) and non-yoga (had not practiced yoga or practiced yoga at least 12 weeks ago).

### 2.3. Measurements

#### 2.3.1. Primary Outcome: Arterial Stiffness

Following standard guidelines [10], pulse wave velocity (PWV) (m/s) was calculated by dividing arterial path length by the pulse transit time between the carotid and superficial femoral arterial segments. Using a calliper, the path length was calculated as the direct distance between the two arterial segments [11]. The pulse transit time was captured using a topometric device (SphygmoCor SCOR-Px, AtCor Medical, Sydney, Australia); specifically, a high-fidelity tonometer that captures sequential pressure waveforms at the carotid and femoral sites. The pressure waveforms were gated using an electrocardiogram and the device calculated the time between the carotid and femoral sites using the foot-to-foot method [12]. Measurements were taken in triplicate and the mean of the closest two were used for data analysis.

#### 2.3.2. Covariates: Demographic

Participants were asked to self-report their age (years/months) and biological sex (female or male). 

#### 2.3.3. Covariates: Hemodynamic 

Mean arterial pressure (MAP, mm Hg) and resting heart rate (RHR, beats.min^−1^) were recorded using an automated device (SphygmoCor XCEL, AtCor Medical Pty. Ltd., Sydney, Australia). Measurements were taken in triplicate and the mean of the closest two were used for data analysis.

#### 2.3.4. Covariates: Cardiometabolic

Waist-to-hip ratio, total cholesterol (mg.dL^−1^) and blood glucose (mg.dL^−1^) were measured. Waist-to-hip ratio (WHR) was measured (nearest 0.1 cm) using a retractable tape measure, following a published protocol [13]. Total cholesterol and blood glucose were measured following an overnight fast (except water) using capillary samples and a portable analyser (CardioChek PA, Whitestown, IN, USA).

#### 2.3.5. Covariates: Lifestyle

Cigarette smoking status, physical activity, sedentary behaviour and perceived stress were recorded. Physical activity levels (MET-min.week^−1^) and time spent sedentary (sitting time, min.week^−1^) were estimated using the International Physical Activity Questionnaire Short Form (IPAQ-SF) (www.ipaq.ki.se, accessed on 1 February 2021). Perceived stress was measured using the 14-term Perceived Stress Scale (PSS) [14]. Smoking status was self-reported.

#### 2.3.6. Sample Size

The sample size was estimated using a freely available spreadsheet [15], where the sample size required to detect the smallest beneficial effect in a group comparison study between a stretch and non-stretch group (in this case, 0.2 of the between-subject standard deviation for pulse wave velocity [0.265 m.s^−1^] found in Nishiwaki et al. (2015) [16]) with the maximum chances of a Type 1 and 2 error set at 5% (i.e., very unlikely), was approximately 272 subjects. To allow for an approximate 40% non-completion rate across all study measures, at least 381 subjects were targeted; however, we accepted 82 more once interest in the study gained momentum and individuals wanted to be involved.

#### 2.3.7. Statistical Analyses

Statistical analyses were performed using Jamovi version 2.3.2.0. Descriptive statistics are presented as means and standard deviations or frequencies and percentages. Comparisons between yoga and non-yoga groups were performed using independent samples *t*-tests and chi-squared tests, as appropriate. Mean differences (MD) are also reported. After testing for violation of assumptions using standard procedures, the difference in carotid–femoral pulse wave velocity (cfPWV) based on yoga participation was assessed using analysis of covariance (ANCOVA). The initial model accounted for known haemodynamic factors that alter cfPWV, including RHR and MAP. Age and biological sex (male or female) were also included in Model 1 due to group differences and known associations with PWV. Next, Model 2 used the cofounding variables from the previous model but included further adjustment for lifestyle-related factors, physical activity, total sedentary behaviour (sitting time min.week^−1^), perceived stress score and smoking. Finally, Model 3 again used the previous confounding variables and included cardiometabolic risk factors that may explain variability in cfPWV, including waist-to-hip ratio, total cholesterol and fasting glucose. Bonferroni post hoc comparisons were performed to determine the simple mean difference between groups and produce 95% confidence intervals. We used a difference of 1 m.s^−1^ in cfPWV between yoga and non-yoga groups to adjudicate clinical significance [1]. Finally, a partial correlation was completed, analysing the association between cfPWV and MAP in yoga participants while controlling for known confounders, including age, sex and RHR.

## 3. Results

### 3.1. Participants

Out of 463 participants who expressed interest (see Figure 1), 383 were eligible for participation, including 202 classified as yoga participants. European New Zealanders were the predominant ethnic group (73.1%), with 11.5% Indian, 8.4% Asian, 3.7% African, 2.3% Māori/Pasifika and 1.0% others also being represented. The yoga group engaged in yoga a mean 2.8 ± 1.6 times/week and were more flexible compared to the non-yoga group (37.1 + 8.4 compared to 24.3 + 11.0 cm (*p* < 0.001) as measured by the sit-and-reach test). As reported in Table 1, the yoga group was older (48.4 ± 14.1 vs. 42.8 ± 14.1) and included more females (81% vs. 44% female). Four of the covariates deemed a priori to confound the outcome were also different, with the yoga group having higher cholesterol (MD: 16 mg.dL^−1^), lower waist-to-hip ratio (−0.1), greater physical activity (862.5 MET-min.week^−1^), less sitting time (−54.7 min.week^−1^) and lower perceived stress (−0.8). 

### 3.2. Arterial Stiffness

Figure 2 descriptively compares cfPWV by yoga participation and age. Table 2 presents the statistical differences for the modelling of cfPWV between yoga and non-groups. The initial model (Model 1) identified a significantly lower cfPWV for the yoga versus non-yoga group when controlling for age, sex, MAP and RHR, and Bonferroni post hoc analysis indicated this difference was −0.29 m.s^−1^, (95% CI = 0.11 to 0.56, *p* = 0.003). The cfPWV remained lower for the yoga versus non-yoga group following adjustment for all covariates, with Bonferroni post hoc analysis finding a −0.28 m.s^−1^ difference between groups (95% CI = −0.55 to 0.08, *p* = 0.008). The correlation between cfPWV and MAP in yoga participants after accounting for age, sex and RHR was 0.50 (*p* < 0.0001).

## 4. Discussion

The aim of this study was to determine the effect of yoga on cfPWV in non-hypertensive, otherwise healthy adults. The principal finding was that adults that partake in regular yoga sessions have significantly lower cfPWV compared to non-yoga participants after controlling for age, sex, MAP and RHR. This finding suggests that undertaking regular yoga may be beneficial for participants’ cardiovascular health.

### 4.1. Limitations and Strengths

Firstly, the generalizability of our findings is limited to self-reported healthy adults and cannot be extended to hypertensive or otherwise unhealthy cohorts. Secondly, we did not collect data on yoga style or intensity, which may influence exercise work rate, and therefore subsequent physiological adaptations and cfPWV changes. A major strength of the study is that this is the first large-scale study to examine the association between cfPWV and yoga participation among a healthy adult population.

### 4.2. Comparison to the Literature

The major finding of this study is that after adjusting for known confounders associated with cfPWV, including age, sex, MAP and RHR, we found significantly lower cfPWV in the yoga compared to the non-yoga groups. Further adjustment for lifestyle (model 2) and cardiometabolic confounders (model 3) had little further effect on the strength of the association. However, the lower cfPWV in the yoga compared to the non-yoga group (−0.29 m.s^−1^) was smaller than the reported clinically meaningful change of 1.0 m.s^−1^ [1]. Previous research using a 12-week controlled trial found that practicing yoga, compared to brisk walking for 1 h per day, 6 days per week, reduced cfPWV by 1.3 m.s^−1^ in hypertensive older adults [4]. Differences in study design, including yoga dosage and participant cohort demographics may account for differences in the magnitude of cfPWV change between this study and that of Patil et al. (2015) [4].

One of the major changes that occurs with practicing yoga is improved flexibility [17]. Indeed, it was evident from the current study that those who regularly practiced yoga were more flexible than the non-yoga group (Table 1). This has led some researchers to suggest that improved flexibility is associated with reduced arterial stiffness [18]. More recent research seems to provide further evidence for this association with healthy adults with high flexibility having better arterial compliance than low-flexibility adults [19]. Furthermore, a number of training studies have found that passive stretching (a major part of the yoga session) is associated with a reduction in arterial stiffness [16,20,21].

Acutely, stretching during yoga results in an alteration of blood flow to the area of the stretch elicited by the two competing responses of vasoconstriction, triggered by an increase in sympathetic nervous stimulation [22] due to feedback from stress on the muscle mechanoreceptors and metaboreceptors, and vasodilation from release of vasoactive substances as a result of stretch-induced stress on the vessels wall [23]. Applying passive stretching over a long period of time would result in a number of adaptations to this system that may result in reduced artery stiffness. For example, the reduced central artery stiffness in the yoga participants may have come about from a reduction in central blood pressure as a result of stimulation of sympathetic nerves controlling cardiac output and thereby PWV [22]. Indeed, after controlling for known pulse wave velocity confounders, including age, sex and resting heart rate, we found a correlation of 0.50 between pulse wave velocity and mean arterial pressure in the yoga participants of this study, indicating the close relationship between blood pressure and arterial stiffness. Alternatively, the long-term passive stretching undertaken by the yoga participants may have reduced artery stiffness by remodelling the arterial wall via an increase in the elastin and collagen content [24]. Future research is required to elucidate the mechanisms involved.

The current study has indicated that the overall amount of clinical benefit from such yoga exercise is small, but from the shape of the graph may be more beneficial for older individuals. In particular, older individuals or individuals who are injured or unable to partake in moderate-to-high-intensity aerobic exercise may therefore benefit from partaking in low-intensity yoga exercise.

## 5. Conclusions

The effect of yoga on arterial stiffness is relatively unknown; therefore, this study aimed to investigate whether lower arterial stiffness was associated with regular yoga practice. The findings from this study indicate that regular yoga exercise was associated with a 0.3 m.s^−1^ reduction in central vascular stiffness compared to non-yoga participants. This represents a useful non-pharmacological treatment for improving vascular health, which may be particularly useful for older individuals and those that cannot undertake traditional high-intensity aerobic exercise.

## Figures and Tables

**Figure 1 ijerph-20-05852-f001:**
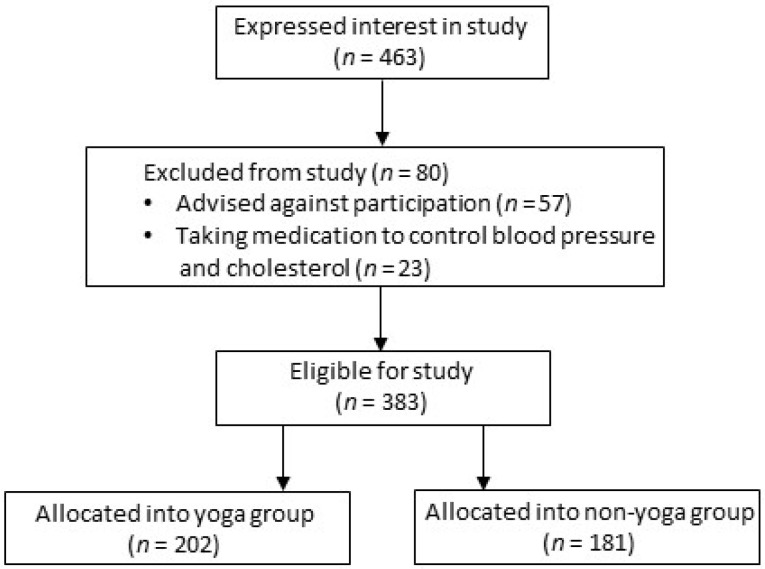
Participant recruitment details.

**Figure 2 ijerph-20-05852-f002:**
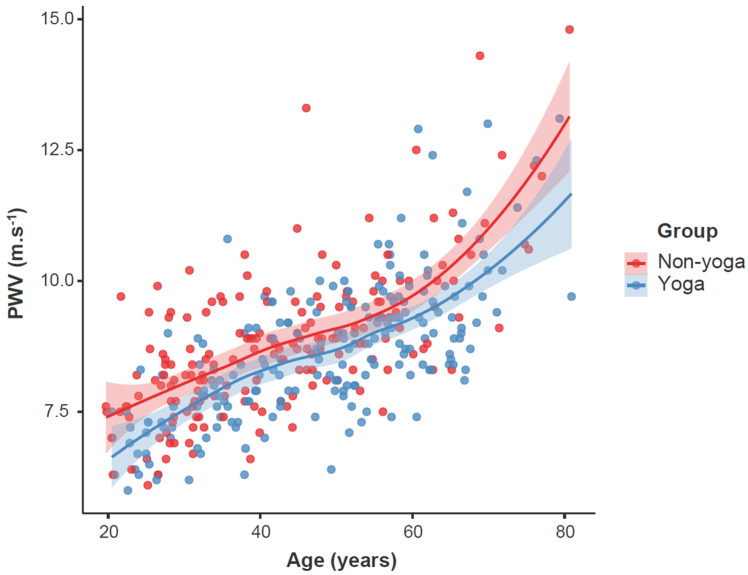
Association between age and carotid–femoral pulse wave velocity (cfPWV) by yoga participation. Data are individual cfPWV values for the non-yoga (red) and yoga (blue) participants. The solid lines are the smoothed regression lines and the shaded areas are the standard errors.

**Table 1 ijerph-20-05852-t001:** Summary of carotid–femoral pulse wave velocity (cfPWV), demographic, hemodynamic, cardiometabolic and lifestyle data by yoga participation.

	Yoga (*n* = 202)	Non-Yoga (*n* = 181)		*p* Value
Continuous Variables	Mean ± SD	Mean ± SD	Mean Diff.	*p*
Yoga participation (sessions.week^−1^)	2.8 ± 1.6	0		
cfPWV (m.s^−1^)	8.7 ± 1.3	8.8 ± 1.4	−0.1	0.330
Age (years)	48.4 ± 14.1	42.8 ± 14.1	5.6	<0.001
Mean arterial pressure (mmHg)	88.1 ± 9.7	88.9 ± 9.0	−0.8	0.410
Resting heart rate (beats.min^−1^)	57.7 ± 8.8	59.1 ± 9.1	−1.4	0.120
Total cholesterol (mg.dL^−1^)	192.2 ± 40.5	176.7 ± 42.8	15.5	<0.001
Blood glucose (mg.dL^−1^)	90.6 ± 13.8	92.0 ± 15.3	−1.4	0.360
Waist-to-hip ratio	0.83 ± 0.1	0.88 ± 0.1	−0.05	<0.001
Physical activity (MET-min.week^−1^)	3446.7 ± 2876.1	2584.2 ± 2558.9	862.5	0.002
Sitting time (min.week^−1^)	331.1 ± 163.6	385.8 ± 181.5	−54.7	0.002
Perceived stress score	27.3 ± 3.3	28.1 ± 3.9	−0.8	0.03
Count Variables	n:n	n:n		*p*
Female:Male	164:38	81:100	-	<0.001
Smoking yes:no	10:192	13:168	-	0.36

Values are means ± SD.

**Table 2 ijerph-20-05852-t002:** Results of the analysis of covariance modelling for carotid–femoral pulse wave velocity (cfPWV) between the yoga and non-yoga groups.

	Mean Diff. (m.s^−1^)	Sum of Squares	df	Mean Square	*p* Value
Model 1	0.29	236.1	5	47.2	<0.001
Model 2	0.26	228.3	9	25.4	<0.001
Model 3	0.28	213.3	12	17.8	<0.001

Model 1 controlled for confounders of cfPWV, including age, sex, resting heart rate and mean arterial pressure. Model 2 included lifestyle factors of physical activity, sedentariness, smoking and perceived stress. Model 3 included cardiometabolic risk factors of waist-to-hip ratio, total cholesterol and blood glucose concentrations.

## Data Availability

The data presented in this study are available on request from the corresponding author. The data are not publicly available due to ethical concerns.

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
