# Peer review of "Association between Yoga Participation and Arterial Stiffness: A Cross-Sectional Study"

_ijerph, 2023, doi:10.3390/ijerph20105852_

Round 1
Reviewer 1 Report
This study analyzed 383 healthy subjects who had routine yoga practice and who did not, and found routine yoga practice lowered cfPWV by ~0.3m/s when several factors (like age/sex/HR..) were controlled.
comments:
1) The authors used SphygmoCor Xcel device to measure cfPWV. This instrument also determines central blood pressure (I think the mean arterial pressure data in Table 1 were from SphygmoCor Xcel, too). However, like cfPWV, the mean arterial pressure in Table 1 also did not show statistical significance between the yoga group and the control group when the statistical analysis did not control for factors such as age and sex. Since the authors suggested in the Discussion that the observed reduced arterial stiffness in the yoga subjects could be due to reduced central BP (line 197-201), the authors should be able to verify this claim by repeat ANCOVA on mean arterial pressure and see whether there will be any association between central BP and cfPWV after controlling the confounding factors.
2) Explain what the shades in Figure 2 are.
3) It’d be clearer for the readers if the different confounding factors in each model are listed in the body of Table 2. Also define df. Table 2 should be explained a bit more in the Results section. Did the three models yield different findings? Line 172-177 should be put in the Results section.
4) Line 160-: In Table 1, there is no statistically significant difference in cfPWV between the two groups. More concisely, this sentence starting line 160 should mention that the statistically significant difference was revealed only after controlling for the demographic factors (age/sex) as in Table 2….
5) Line 182 ref;
6) Line 169: This statement is not accurate. Similar studies have been reported (e.g., PMID 18226247 & 24149206). But perhaps this is the first “large-scale” study of such kind.
7) Line 165: “self-reported” healthy adults. Unless you have criteria for determining healthy people.
8) Line 146: MAP and RHR should be spelled out upon their first appearance in the paper.
Author Response
- The authors used SphygmoCor Xcel device to measure cfPWV. This instrument also determines central blood pressure (I think the mean arterial pressure data in Table 1 were from SphygmoCor Xcel, too). However, like cfPWV, the mean arterial pressure in Table 1 also did not show statistical significance between the yoga group and the control group when the statistical analysis did not control for factors such as age and sex. Since the authors suggested in the Discussion that the observed reduced arterial stiffness in the yoga subjects could be due to reduced central BP (line 197-201), the authors should be able to verify this claim by repeat ANCOVA on mean arterial pressure and see whether there will be any association between central BP and cfPWV after controlling the confounding factors.
Q1 Reply. Thank you for this comment. We have now completed a partial correlation between mean arterial blood pressure and pulse wave velocity in the yoga group and after controlling for known pulse wave velocity confounders including age, sex and resting heart rate and we found a correlation of 0.50 suggesting that reduced arterial stiffness is indeed associated with reduced mean arterial pressure. We have inserted this information into the Discussion text (see Line 200-203)
2) Explain what the shades in Figure 2 are.
Q2 Reply. . We apologise for this error. The details of this figure were inserted into the text below the figure. We have since moved them to the legend of the figure.
3) It’d be clearer for the readers if the different confounding factors in each model are listed in the body of Table 2. Also define df. Table 2 should be explained a bit more in the Results section. Did the three models yield different findings? Line 172-177 should be put in the Results section.
Q3Reply. . Thank you for this comment. The confounding factors are already listed in this Table for the 3 models used. Also, the results on the variable we were interested in (i.e., pulse wave velocity between the yoga and non-yoga groups) are also listed in the table. As indicated the three models did not yield different results (see the p values all the same) and very close differences in pulse wave velocities between the models were observed. We feel this information is clear enough for readers to understand without other detail added. No further change was made here apart from the fact that we inserted ‘degrees of freedom’ into the table for clarity.
4) Line 160-: In Table 1, there is no statistically significant difference in cfPWV between the two groups. More concisely, this sentence starting line 160 should mention that the statistically significant difference was revealed only after controlling for the demographic factors (age/sex) as in Table 2….
Q4Reply. . Thank you for this comment. We have altered the text in this line accordingly.
5) Line 182 ref;
Q5Reply. . Thank you we have added this reference.
6) Line 169: This statement is not accurate. Similar studies have been reported (e.g., PMID 18226247 & 24149206). But perhaps this is the first “large-scale” study of such kind.
Q6Reply. . Thank you for this comment. Yes, the previous studies mentioned were small scale (n < 30) and one was on females only, so we are happy to insert in the text “large-scale” study.
7) Line 165: “self-reported” healthy adults. Unless you have criteria for determining healthy people.
Q7Reply. . Agree, should be “self-reported” healthy adults. We have changed the text accordingly.
8) Line 146: MAP and RHR should be spelled out upon their first appearance in the paper.
Q8Reply. . They have been. See line 82.
Reviewer 2 Report
Raj_Yoga and Arterial_ijerph_2023
Thank you for the opportunity to review this manuscript. The paper is original, it is well structured overall, it was found that it adds to current knowledge, and it is of importance to clinicians and patients.
However, I have some suggestions highlighted bellow.
Specific comments
Introduction.
Line 32. Can you explain a little more about the mechanism in which Yoga works to ameliorate arterial stiffness?
2.2. PARTICIPANTS.
Line 65: “non-yoga (less than once/week for < 12 weeks)” are you sure about this criteria? I suppose it should be also: ≥ 12 weeks. If not maybe, for example, they might have been practicing yoga for at least once/week, three weeks ago.
2.3.1. PRIMARY OUTCOME: ARTERIAL STIFFNESS
Line 69. Please include here as the first time in the main text, the explanation of the acronym PWV: pulse wave velocity.
2.4. SAMPLE SIZE
Line 103. Please state the name of the author and after the number of the reference in reference 13.
What was the expected difference between groups in pulse wave velocity used to calculate the sample size?
2.5. STATISTICAL ANALYSES
Line 114. Please include here as the first time in the main text, the explanation of the acronym cfPWV.
The models 2 and 3, they are supposed to include the factors tested in the previous models? Please clarify. It looks like this but it is not clear.
3. Results
Line 153-155: “Data are individual cfPWV values for the non-yoga (red) and yoga (blue) participants. The solid lines are the smoothed regression lines and the shaded areas are thestandard errors.” Please, include this text in the legend of the figure 2.
Table 2. to what statistic or variable, the p values in the table 2 refer (the three < 0.001)? In the first time, it might be thought that they refer to cfPWV differences, but in the text p values are different. Are the p values in the text referred to post-hoc comparison and the p values in the table to the between-subject effect analysis? Please clarify.
Maybe it is interesting to add some information about the models? For example, which of the covariates or factors showed also significance in the between-subject effect analysis?
4. Discussion
Line 161, after this “non-yoga participants”, please add: “after adjusting for potential confounding factors”.
Line 182. Please, add reference, at the end of the sentence.
Line 184. “Indeed, it was evident from the current study that those who regularly practiced yoga were more flexible than the non-yoga group (table 1)." In which data from the table 1 do you base to affirm that the yoga group was more flexible. I think it is not correct. You can say that they are supposed to be more flexible according with the data presented in the table 1 (levels of activity and lifestyle) but nothing more.
I think the English is correct and the manuscript reads well.
Author Response
Specific comments
Introduction.
Line 32. Can you explain a little more about the mechanism in which Yoga works to ameliorate arterial stiffness?
Q2.1 Reply. Thank you for this suggestion. We have added the following text to the introduction “It is thought that yoga itself (through controlled breathing and relaxation) enables a reduction in the sympathetic innervation of the smooth muscle at the blood vessel (Bhaskar et al 2017), while others have postulated that regular stretching that occurs with yoga may cause increased release and subsequent enhanced bioavailability of nitric oxide causing endothelium-dependent decreased vasoconstrictor tone (Hotta et al 2018)”.
2.2. PARTICIPANTS.
Line 65: “non-yoga (less than once/week for < 12 weeks)” are you sure about this criteria? I suppose it should be also: ≥ 12 weeks. If not maybe, for example, they might have been practicing yoga for at least once/week, three weeks ago.
Q2.1Reply. Thank you for this observation. We agree its confusing. We have altered the text to make this clear. “Participants were separated into yoga (practiced for at least once/week for ≥ 12 weeks) and non-yoga (have not practiced yoga or practiced yoga at least 12 weeks ago).”
2.3.1. PRIMARY OUTCOME: ARTERIAL STIFFNESS
Line 69. Please include here as the first time in the main text, the explanation of the acronym PWV: pulse wave velocity.
Q2.3.1Reply. Thank you we have altered the text accordingly.
2.4. SAMPLE SIZE
Line 103. Please state the name of the author and after the number of the reference in reference 13.
What was the expected difference between groups in pulse wave velocity used to calculate the sample size?
Q2.4Reply. Thank you for this comment. We have inserted the authors name in the text. The difference in PWV between groups to calculate sample size was 0.265 m/s.
2.5. STATISTICAL ANALYSES
Line 114. Please include here as the first time in the main text, the explanation of the acronym cfPWV.
The models 2 and 3, they are supposed to include the factors tested in the previous models? Please clarify. It looks like this but it is not clear.
Q2.5Reply. We agree and have inserted carotid-femoral pulse wave velocity (cfPWV).
Indeed, the models included all the confounding factors from previous models. We have inserted this in the text to make this clear.
- Results
Line 153-155: “Data are individual cfPWV values for the non-yoga (red) and yoga (blue) participants. The solid lines are the smoothed regression lines and the shaded areas are thestandard errors.” Please, include this text in the legend of the figure 2.
Q3.Reply. We have removed this from the text and added it to the figure legend/caption.
Table 2. to what statistic or variable, the p values in the table 2 refer (the three < 0.001)? In the first time, it might be thought that they refer to cfPWV differences, but in the text p values are different. Are the p values in the text referred to post-hoc comparison and the p values in the table to the between-subject effect analysis? Please clarify.
Maybe it is interesting to add some information about the models? For example, which of the covariates or factors showed also significance in the between-subject effect analysis?
Reply.Thank you for your comments. The p values in the tables are for the overall significance of the models between the yoga and non-yoga groups. The p values in the text refer to the Bonferroni post hoc analysis comparisons to get mean differences and 95% confidence limits. We have added this information to the text. (see results section)
We are reluctant to examine the models in too much depth as this not only confuses the issues but breaking the models down into separate variables is in our opinion unhelpful and not statistically wise. We have therefore not done this.
- Discussion
Line 161, after this “non-yoga participants”, please add: “after adjusting for potential confounding factors”.
Reply.Thank you for this comment, we have added this text to the sentence.
Line 182. Please, add reference, at the end of the sentence.
Reply.Thank you we have added the reference number to this sentence.
Line 184. “Indeed, it was evident from the current study that those who regularly practiced yoga were more flexible than the non-yoga group (table 1)." In which data from the table 1 do you base to affirm that the yoga group was more flexible. I think it is not correct. You can say that they are supposed to be more flexible according with the data presented in the table 1 (levels of activity and lifestyle) but nothing more.
Reply.Thank you for this observation. We have subsequently added this information (e.g. flexibility between the groups) into the start of the results section.
Round 2
Reviewer 2 Report
I want to thank the authors the effort made to ameliorate the manuscript. I have only some minor corrections suggested bellow.
Please include the difference in PWV between groups (0.265 m/s) used to calculate sample size in the sample size section.
Please, add the sit and reach test, in the material and methods section due to the fact that you have included the results of the test, in the results section at the moment.
According to the suggestion of the other reviewer, you have inserted the next sentence in the discussion section: “Indeed, after controlling for known pulse wave velocity confounders including age, sex and resting heart rate, we found a correlation of 0.50 between pulse wave velocity and mean arterial pressure in the yoga participants of this study indicating the close relationship between blood pressure and arterial stiffness”. I think that you have the adequate information about this analysis in the statistical analyses section and you have to present the result also, previously in the results section.
Author Response
I want to thank the authors the effort made to ameliorate the manuscript. I have only some minor corrections suggested bellow.
My Reply: Thank you for your suggestions. Please find my comments and changes in the manuscript in blue ink (track changed).
Please include the difference in PWV between groups (0.265 m/s) used to calculate sample size in the sample size section.
My Reply: Thank you for this suggestion, we have made this change.
Please, add the sit and reach test, in the material and methods section due to the fact that you have included the results of the test, in the results section at the moment.
My Reply: Thank you for this suggestion, we have made this change.
According to the suggestion of the other reviewer, you have inserted the next sentence in the discussion section: “Indeed, after controlling for known pulse wave velocity confounders including age, sex and resting heart rate, we found a correlation of 0.50 between pulse wave velocity and mean arterial pressure in the yoga participants of this study indicating the close relationship between blood pressure and arterial stiffness”.
I think that you have the adequate information about this analysis in the statistical analyses section and you have to present the result also, previously in the results section.
My Reply: Thank you for this suggestion, we have added the information into the statistical analysis section and results section.